# The Effect of Flushing on the Nitrate Content and Postharvest Quality of Lettuce (*Lactuca sativa* L. Var. *Acephala*) and Rocket (*Eruca sativa* Mill.) Grown in a Vertical Farm

Davide Guffanti [1], Giacomo Cocetta [1,*], Benjamin M. Franchetti [2] and Antonio Ferrante [1]

1   Department of Agricultural and Environmental Sciences, Università degli Studi di Milano, Via Celoria 2, 20133 Milan, Italy; davide.guffanti@unimi.it (D.G.); antonio.ferrante@unimi.it (A.F.)
2   Agricola Moderna, Sunspring Srl, Via S. Pertini 26, 20066 Melzo, Italy; benjamin@agricolamoderna.com
*   Correspondence: giacomo.cocetta@unimi.it; Tel.: +39-02-5031-16612

**Abstract:** Hydroponics is the most widely used technique in closed cultivation environments, and this system is often used for the cultivation of baby leaf vegetables. These species can accumulate high levels of nitrates; for this reason, the control of growing conditions is a crucial factor for limiting their content, especially in protected cultivations. The aim of this work was to reduce nitrate accumulation in leafy vegetables grown in a vertical farm while preserving the quality at harvest as well as during storage. This objective was achieved by completely replacing the nutrient solution with water a few hours before harvest ("flushing"). The trials were carried out on lettuce (*Lactuca sativa* L. Var. *Acephala*, cv. Greenet) and rocket (*Eruca sativa* Mill., cv. Rome). Three independent trials were conducted on lettuce, applying the flushing treatment 24 h and 48 h prior to harvest. One trial was conducted on rocket, applying the treatment 48 h before harvesting. Sampling and related analyses were carried out at harvest and during the storage period to determine chlorophyll, leaf fluorescence, total sugars, chlorophyll ($a + b$), carotenoids, phenolic index, anthocyanins and nitrate content. Moreover, relative humidity (RH%), $O_2$% and $CO_2$% determination inside the package headspace were monitored during storage. The results obtained indicate that it is possible to reduce the nitrate concentration by up to 56% in lettuce and 61% in rocket while maintaining the product quality of baby leaves by replacing the nutrient solution with tap water before harvest.

**Keywords:** baby leaves; hydroponics; lettuce; nitrate; rocket

## 1. Introduction

Vertical farming is an indoor cultivation system introduced as a technical solution in which the crop cultivation area is extended in the vertical dimension to increase production capacity per cultivated surface area [1]. Vertical farms are often located in urban or peri-urban areas where arable land surfaces are limited. This solution includes the adoption of soilless cultivation systems, in which soil is replaced by water and/or a substrate that can provide support and mineral elements to the plant. These production technologies contribute to the simplification of different procedures, including harvesting. Moreover, proper management of the nutrient solution can help to improve the product quality and to improve the economic and environmental sustainability of the production [2]. Hydroponics is the technique mostly widely used in vertical farming [3]. Plant roots can be immersed in liquid solutions containing macro- and micronutrients, provided in proper concentrations to guarantee good yield and quality. Growing substrates, such as peat, perlite, vermiculite, rock wool and zeolite, are used as soil substitutes to provide support to the roots [4]. This solution provides several advantages compared to traditional soil cultivation, including: the elimination of or reduction in ground-related problems (insects, fungi and bacterial contamination), the increased productivity of the plant growing process, a more rational use of water, fertilizers and pesticides, a reduction in agronomic operations (weeding,

tillage, etc.), a reduction in the working hours necessary to manage larger productive areas and, finally, a more efficient way to control the electrical conductivity (EC) and pH levels [5]. This system is often applied in indoor cultivation environments to provide protection and maintain optimal growth conditions during the whole crop cycle. Under these conditions, it is possible to manage physical and environmental factors such as light (quality and quantity), temperature, humidity, $CO_2$ concentration, air recirculation and oxygenation of the rhizosphere [6].

Lettuce (*Lactuca sativa* L.) and rocket (*Eruca sativa* Mill.) are among the most widespread baby leaf vegetable species grown in vertical farming [1]. Baby leaves are appreciated by consumers and often used in the fresh-cut market. These kinds of vegetables are a good source of fiber, vitamins and antioxidant molecules [7]. The quality of leafy vegetables, including lettuce and rocket, can be defined by different parameters, which together contribute to the commercial success of these products. Sugars play an important role in plant physiology, as they are the main product resulting from photosynthetic activity. Sugars also contribute to the sensory properties of leafy vegetables and are very important in the maintenance of quality during shelf life and storage [8]. Color is a key aspect guiding consumers when they choose a product in a retail shop. In the case of leafy greens, chlorophyll is the determinant of this characteristic. At the same time, chlorophyll content is an important indicator of plant health status and of potential photosynthetic activity [9]. Leaf functionality and the potentiality of photosystems (mainly photosystem II, PS II) can be evaluated non-destructively by measuring chlorophyll a fluorescence and its related parameters. In recent years, analyses of chlorophyll a fluorescence parameters have been successfully used for estimating the quality of leafy greens, including during storage [10]. Lettuce and rocket are also considered moderate sources of important antioxidant and health-related molecules, which can provide important traits related to the nutritional quality of these foods. These compounds include phenolics, anthocyanins and carotenoids, among others [8–11]. However, most of the species grown as baby leaves, including lettuce and rocket, tend to accumulate large amounts of nitrate ($NO_3$) in vacuoles. Nitrate is a non-toxic molecule; however, the higher risk for humans is related to the nitrate reduction to nitrite in the mouth and stomach. Nitrite can react with other substances such as amines or amino acids contained in food, leading to nitrosamine formation, some of which are considered carcinogenic substances, and others are considered substances with direct mutagenic effects. In the circulatory system, nitrates can convert hemoglobin into methemoglobin, reducing oxygen and causing the so-called "blue baby syndrome" or "cyanotic baby syndrome" in newborns [12]. A significant accumulation of nitrates is often observed in plant tissues, such as leaves and stems, due to insufficient nitrate reductase (NR) enzyme activity. Plants can absorb nitrogen (N) as nitrate ($NO_3^-$) or ammonium ($NH_4^+$), but $NO_3$ is the most available nitrogen form in soil. NR is the enzyme responsible for the first step in the conversion of $NO_3$ into organic nitrogen. Nitrate fertilization usually allows higher yields of crops to be obtained compared to ammonium [13]. The plant's ability to accumulate nitrate depends on its genotypic background and on environmental factors.

There are several factors promoting the accumulation of nitrates in a protected environment cultivation system, including low light availability, light quality variation, high water availability, high temperatures, pH, aeration management (high $CO_2$ level) and nutrient solution quality. Fertigation management is a common factor playing an important role in controlling nitrate accumulation. Reducing the EC of the nutrient solution by adding water or completely substituting the nutrient solution with water in the last days before harvest can help in reducing the amount of nitrate in leaves [14]. However, this operation could negatively affect the plant growth rate [15] and negatively affect the final product quality. Nowadays, it is possible to adopt effective lighting management strategies to reduce the nitrate level. The increase in photosynthetic activity can have an important role in reducing nitrate accumulation [16]. NR activity is affected by the light spectrum, light intensity and photoperiod. Red light has been shown to stimulate nitrate reductase activity, leading to a reduction in nitrate concentration [17]. The photoperiod is especially important in

short-day plants because nitrate is accumulated mainly at night, when its accumulation in vacuoles is more efficient. It has been shown that a continuous red-light treatment one day before harvest will successfully reduce the assimilation of nitrates [18]. However, plant development and growth under LED lighting are highly dependent on the whole environment and growth system, including temperature and mineral nutrient availability.

Green leafy vegetables such as lettuce or rocket can accumulate an excessive amount of nitrate, constituting a serious problem for consumers [19]. However, today, with the introduction of soilless cultivation systems, greater control of the content of nitrates as well as phytochemical compounds and nutrients has been achieved [20].

The maximum nitrate levels reported in EU Regulation No. 258/2011 are 4000–5000 mg $NO_3$/kg for *Lactuca sativa*, cultivated in a protected environment, and 6000–7000 mg $NO_3$/kg for *Eruca sativa* [21].

Once harvested, during storage, transformation and shelf life, the product undergoes fluctuations in the nitrate content, and this can be affected by different environmental conditions. The higher risk during storage is the accumulation of nitrite deriving from the nitrate present at harvest. This phenomenon can be promoted during leafy vegetable storage by the low $O_2$ concentration present in the package headspace when modified atmosphere packaging (MAP) is applied [22]. MAP allows the product shelf life to be extended: the oxygen reduction and the carbon dioxide increase reduce tissue respiration, ethylene production, enzymatic reactions and some physiological alterations, thus contributing to longer-lasting quality [23]. The aim is to obtain an optimal composition, which is passively created by the product itself, thanks to the respiratory activity, or actively by injecting an appropriate gas mixture before closing the packaging. The gas composition within the package is affected by the metabolic activity of the product as well as the properties of the plastic film used [24]. The tolerance to low oxygen and high carbon dioxide concentrations depends on the product characteristics. In the case of non-optimal storage conditions, product spoilage can occur, with the accumulation of undesired compounds that lead to physiological disorders and loss of sensory quality [25].

The aim of the present study was to develop a strategy for reducing the nitrate content in lettuce and rocket grown in a vertical farm through the management of the nutrient solution. This was achieved by substituting the nutrient solution with water a few hours before harvesting. The individuation of the optimal moment for applying the treatment was crucial for reducing nitrate content while maintaining the yield and preserving the produce quality. The quality retained was also evaluated during postharvest, simulating the normal commercial management of fresh-cut products.

## 2. Materials and Methods

### 2.1. Experimental Set-Up

The experiments were conducted in an indoor vertical farm system of a commercial company that cultivates baby leaves. The trials were carried out on lettuce (*Lactuca sativa* L. Var. *Acephala*, cv. Greenet) and rocket (*Eruca sativa* Mill., cv. Rome) grown indoors in a hydroponic ebb-and-flow system. Plants were sown in trays containing a plug filled with a peaty substrate (pH 5.5–5.9) and grown with a photoperiod of 16 h, an ambient temperature of 21–22 °C and a relative humidity of 60–70%. The illumination was provided with LED (28 W PAR LED lamps, Valoya, Helsinki, Finland) lights, and the light intensity was 170 μmol m$^{-2}$ s$^{-1}$. The nutrient solution consisted of modified Hoagland's solution: the nutrient concentrations, expressed as mM, were: 12 N-$NO_3$, 3.8 N-$NH_4$, 2.8 P, 8.4 K, 3.5 Ca, 1.4 Mg, 3.5 S$^-$ and Hoagland's concentrations for micronutrients. In order to describe a real productive context, tap water was used in the preparation of the nutrient solution.

The main objective of this work was achieved by completely replacing the nutrient solution with tap water a few hours before harvest. This treatment is called "flushing". Three independent trials were conducted for lettuce.

In the first trial, the flushing treatment was applied twice, 24 h before harvesting with an interval of 12 h. In the second trial (presented in Supplementary Material) and in the

third one, flushing was applied four times, 48 h before harvest at intervals of 12 h. For all the trials, the nutrient solution was applied between flushing treatments.

Based on the previous results in lettuce, one trial was conducted for rocket, in which the flushing treatment was applied four times, 48 h before harvesting at 12 h intervals. All of the analytical determinations were conducted on randomly selected leaves, chosen among 128 plants in the first trial, 144 plants in the second one, 288 plants in the third one and, lastly, 144 rocket plants, for both treatments (flushing and control). Plants were harvested when leaves were around 10–12 cm in height and after harvest, and lettuce and rocket leaf samples (quadruplicates) were collected and stored at $-20\ °C$ until analyzed. The remaining parts of leaves were placed in polypropylene (PP) punnets (~50 g for lettuce, ~25 g for rocket salad) and stored in a cold room at $8 \pm 1\ °C$, with relative humidity at $65 \pm 2\%$, for up to nine days for the storage trial. Four punnets for each time point and treatment were prepared. Sampling and related analyses were carried out at harvest (T0) as well as after two (rocket) or three (lettuce), six and nine days of cold storage. The analyses were carried out at each time point and included evaluations performed in vivo with non-destructive tools and analytical determination of several quality and physiological indexes (destructive methods). Moreover, relative humidity (RH%), $O_2\%$ and $CO_2\%$ levels inside the package headspace were monitored during storage.

### 2.2. Evaluations Performed In Vivo with Non-Destructive Tools

2.2.1. Chlorophyll a Fluorescence

Fluorescence was measured in vivo using the Handy Plant Efficiency Analyzer (PEA, Hansatech, UK) portable fluorometer. From 6 to 12 leaves were randomly analyzed for each treatment and each time point, and they were adapted to the dark for 30 min using leaf clips with a 4 mm diameter. A 3000 $\mu$mol m $^{-2}$ s $^{-1}$ (600 W m $^{-2}$) light pulse was administered to the leaf to measure the fluorescence re-emitted by it. A JIP analysis was performed to calculate the various parameters, such as the performance index (PI), which reflects the photosystem functionality and provides quantitative information on the plant's current performance state, showing any stress conditions [26]. Another parameter considered was the maximum quantum efficiency of photosystem II (Fv/Fm), which is defined as the ratio between variable and maximum fluorescence, and it indicates the probability that an electron captured by the antenna reaches the reaction center by reducing the acceptor. This parameter has a value equal to or greater than 0.83 in herbaceous plant leaves considered healthy.

2.2.2. Non-Destructive Chlorophyll Content Estimation

The chlorophyll content was estimated in vivo through a chlorophyll-meter (CL-01, Hansatech, UK), which provides a leaf greenness indication based on two wavelengths (620 nm and 940 nm). From 6 to 12 leaves were randomly analyzed for each treatment and each time point.

### 2.3. Evaluations Performed at Harvest and during Storage with Laboratory Analyses

2.3.1. Total Sugar Content

Lettuce and rocket leaves were randomly collected for each treatment and each time point. A total of 1 g of leaf tissue (around 2 leaves) was ground in 3 mL and 4 mL of distilled water for lettuce and rocket, respectively. The homogenate obtained was centrifuged at 4000 rpm for 15 min. The total sugar content was then determined in the extract obtained by the anthrone method. The reagent was prepared by diluting 0.1 g of anthrone in 50 mL of sulfuric acid ($H_2SO_4$) and was left stirring under a hood for about 30–40 min until complete clarification. Then, 1 mL of anthrone reagent was added to 0.2 mL of extract placed in Eppendorf tubes, resulting in a blue and a yellow phase. The tubes were placed on ice for 5 min and then vigorously shaken to break the phases. They were heated for 5 min at 95 °C in the Dubnoff bath and finally cooled in water for 5 min. The absorbances were read using a spectrophotometer at a wavelength of 620 nm, and the total sugar content

was calculated based on a calibration line obtained with increasing glucose concentrations ($C_6H_{12}O_6$ 0-0.25-0.5-0.75-1.0-1.5-2.0 mM) [27].

### 2.3.2. Phenolic Index and Total Anthocyanins

The phenolic index was determined by a spectrophotometer directly measuring the leaf extract absorbance [28]. About 30–50 mg of fresh leaf tissue was weighed in 5 mm diameter rods (around 3), and 3 mL of 1% HCl solution in methanol was added. After 24 h of incubation at 4 °C in the dark, the supernatant was read at 320 nm. Values were expressed as ABS 320 nm/g FW. The anthocyanin content was determined by a spectrophotometer on the same extracts. The spectrophotometric reading was performed at 535 nm, and the total anthocyanin content was expressed as 3-glucoside cyanidin equivalents using the molar extinction coefficient: $\varepsilon$ = 29,600 $mM^{-1}$ $cm^{-1}$ [29].

### 2.3.3. Chlorophyll (*a* + *b*) Content and Total Carotenoids

The chlorophyll and carotenoids extraction was carried out using 99.9% methanol ($CH_3OH$). About 30–50 mg of fresh leaf tissue was weighed in 5 mm diameter rods (around 3), and 5 mL of methanol was added. After one night of incubation at 4 °C in the dark, the supernatant was read with the spectrophotometer at 665.2 nm for chlorophyll *a*, 652.4 nm for chlorophyll *b* and 470 nm for total carotenoids. Pigment concentrations were calculated using Lichtenthaler's formulas [30].

### 2.3.4. Nitrate Concentration

The nitrate concentration was determined in leaves with the salicyl-sulfuric acid method [31]. Leaves were randomly collected for each treatment and each time point. For each sample (around 2 leaves), 1 g of leaf tissue was ground respectively in 3 mL and 4 mL of distilled water for lettuce and rocket salad. The extract was centrifuged at 4000 rpm for 15 min, and the supernatant was used for the spectrophotometric determination. For each sample, 80 μL of 5% salicylic acid in sulfuric acid and 3 mL of NaOH (1.5 N) were added to 20 μL of extract. Then, the samples were allowed to cool at room temperature, and readings were taken at 410 nm. The nitrate content was determined using a calibration line prepared with a standard solution of potassium nitrate ($KNO_3$ 0–2.5–5–7.5–10 mM).

### *2.4. Gas Concentration Determination in the Headspace during Storage*

The F-950 Three Gas Analyzer (Felix Instruments, Camas, WA 98607, USA) was chosen for gas analysis inside closed packages. It allows simple and rapid analyses to be carried out, as it is equipped with a suction and delivery system equipped with a needle, which conveys the air from the package to the analyzer and an outlet system, also equipped with a needle inserted in the same way in the package during the analysis. This instrument uses an electrochemical cell to measure the gas concentration (oxygen and carbon dioxide) in the air at a capacity of 200 ppm. It records the date, time, relative humidity and temperature of each sample.

### *2.5. Statistical Analysis*

The data were analyzed using variance analysis (ANOVA) and *t*-test. In the case of a significant *p*-value, the means were compared using Sidak's test. GraphPad Prism was used as the software for statistical analysis. Additional information is reported in the figure and table legends. Detailed results related to the ANOVA analyses are reported in Supplementary Table S1.

## 3. Results

### 3.1. Evaluations Performed In Vivo Using Non-Destructive Tools

3.1.1. Chlorophyll a Fluorescence

The chlorophyll *a* fluorescence data were not significantly affected by the cultivation treatments (Tables 1–3), and no significant changes were observed at harvest for lettuce or rocket.

**Table 1.** Quantitative and qualitative parameters measured at the harvest of treated and untreated *Lactuca sativa* samples.

| | Flushed 24 h | | Flushed 48 h | |
|---|---|---|---|---|
| Treatment | Control | Flushed | Control | Flushed |
| Fv/Fm | $0.83 \pm 0.005$ | $0.82 \pm 0.004$ | $0.84 \pm 0.003$ | $0.84 \pm 0.002$ |
| PI | $0.93 \pm 0.09$ | $0.79 \pm 0.10$ | $1.17 \pm 0.10$ | $1.22 \pm 0.06$ |
| Chlorophyll (AU) | $7.52 \pm 0.44$ | $8.15 \pm 0.05$ | $7.32 \pm 0.31$ a | $6.20 \pm 0.27$ b |
| Total sugar (mg/g Glu eq.) | $15.42 \pm 1.37$ | $12.54 \pm 2.70$ | $10.04 \pm 1.74$ | $14.83 \pm 2.67$ |
| Phenolic index (ABS 320 nm/g) | $19.46 \pm 1.92$ | $21.68 \pm 0.86$ | $23.04 \pm 3.09$ | $31.11 \pm 1.96$ |
| Anthocyanins (mg/100 g) | $20.02 \pm 0.84$ | $21.67 \pm 0.83$ | $18.99 \pm 0.20$ | $18.43 \pm 0.39$ |
| Total chlorophyll $a + b$ (µg/mg) | $1.30 \pm 0.03$ | $1.22 \pm 0.06$ | $1.05 \pm 0.01$ | $1.03 \pm 0.02$ |
| Total carotenoids (µg/mg) | $0.23 \pm 0.008$ | $0.23 \pm 0.01$ | $0.20 \pm 0.001$ | $0.18 \pm 0.005$ |
| Nitrate (mg/Kg) | $1480.47 \pm 100.77$ a | $1093.15 \pm 72.51$ b | $835.75 \pm 99.75$ a | $367.79 \pm 16.16$ b |

Numbers represent total sugar content (mg/Kg Glu eq.), phenolic index (ABS 320 nm/g), anthocyanins (mg/100 g), total chlorophyll $a + b$ (µg/mg), total carotenoids (µg/mg), nitrate content (mg/Kg) (n = 4), chlorophyll fluorescence (Fv/Fm, PI) and chlorophyll (AU) (n = 12). All data are presented as mean $\pm$ SE. When present, different letters represent significant differences ($p \leq 0.05$) among treatments, calculated from *t*-test.

**Table 2.** Quantitative and qualitative parameters measured at harvest and during storage at three, six and nine days after the harvest of treated (flushing 48 h) and untreated *Lactuca sativa* samples.

| Time | $T_0$ | | $T_3$ | | $T_6$ | | $T_9$ | |
|---|---|---|---|---|---|---|---|---|
| Treatment | Control | Flushed | Control | Flushed | Control | Flushed | Control | Flushed |
| Fv/Fm | $0.83 \pm 0.003$ | $0.83 \pm 0.004$ | $0.83 \pm 0.003$ | $0.82 \pm 0.01$ | $0.81 \pm 0.01$ | $0.84 \pm 0.002$ | $0.83 \pm 0.003$ | $0.83 \pm 0.003$ |
| PI | $0.83 \pm 0.08$ a | $0.96 \pm 0.06$ ab | $1.39 \pm 0.11$ ab | $1.12 \pm 0.20$ ab | $1.31 \pm 0.18$ ab | $1.54 \pm 0.07$ b | $1.48 \pm 0.15$ b | $1.53 \pm 0.05$ b |
| Chlorophyll (AU) | $7.69 \pm 0.40$ | $7.92 \pm 0.60$ | $6.62 \pm 0.30$ | $7.73 \pm 0.87$ | $6.98 \pm 0.86$ | $6.55 \pm 0.45$ | $6.22 \pm 0.25$ | $7.42 \pm 0.80$ |
| Total sugar (mg/g Glu eq.) | $4.97 \pm 1.34$ | $4.73 \pm 1.09$ | $3.01 \pm 0.25$ | $4.93 \pm 1.17$ | $3.33 \pm 0.60$ | $4.17 \pm 0.58$ | $2.75 \pm 0.28$ | $3.18 \pm 0.68$ |
| Phenolic index (ABS 320 nm/g) | $16.29 \pm 1.23$ | $14.01 \pm 1.93$ | $18.04 \pm 0.78$ | $16.66 \pm 1.05$ | $17.54 \pm 0.54$ | $15.66 \pm 0.77$ | $15.83 \pm 0.98$ | $18.19 \pm 0.90$ |
| Anthocyanins (mg/100 g) | $20.22 \pm 1.13$ | $17.75 \pm 1.77$ | $26.59 \pm 1.51$ | $22.75 \pm 1.02$ | $25.90 \pm 0.95$ | $23.56 \pm 0.97$ | $21.96 \pm 1.32$ | $24.73 \pm 1.50$ |
| Total chlorophyll $a + b$ (µg/mg) | $0.92 \pm 0.10$ | $1.04 \pm 0.01$ | $1.13 \pm 0.05$ | $1.04 \pm 0.02$ | $1.15 \pm 0.05$ | $1.11 \pm 0.06$ | $1.15 \pm 0.02$ | $1.17 \pm 0.05$ |
| Total carotenoids (µg/mg) | $0.19 \pm 0.01$ | $0.20 \pm 0.007$ | $0.23 \pm 0.006$ | $0.21 \pm 0.005$ | $0.23 \pm 0.01$ | $0.23 \pm 0.01$ | $0.22 \pm 0.003$ | $0.23 \pm 0.008$ |

Numbers represent total sugar content (mg/Kg Glu eq.), phenolic index (ABS 320 nm/g), anthocyanins (mg/100 g), total chlorophyll $a + b$ (µg/mg), total carotenoids (µg/mg), chlorophyll fluorescence (Fv/Fm, PI) and chlorophyll (AU) (n = 8). All data are presented as mean $\pm$ SE. When present, different letters represent significant differences ($p \leq 0.05$) among treatments and time points, calculated from a two-way ANOVA followed by Sidak's test.

**Table 3.** Quantitative and qualitative parameters measured at harvest and during storage at three, six and nine days after the harvest of treated (flushing 48 h) and untreated *Eruca sativa* samples.

| Time | $T_0$ | | $T_2$ | | $T_6$ | | $T_9$ | |
|---|---|---|---|---|---|---|---|---|
| Treatment | Control | Flushed | Control | Flushed | Control | Flushed | Control | Flushed |
| Fv/Fm | $0.85 \pm 0.003$ b | $0.86 \pm 0.001$ b | $0.84 \pm 0.004$ ab | $0.84 \pm 0.003$ ab | $0.82 \pm 0.003$ ab | $0.82 \pm 0.006$ ab | $0.82 \pm 0.004$ ab | $0.81 \pm 0.02$ a |
| PI | $3.86 \pm 0.44$ | $4.13 \pm 0.28$ | $3.82 \pm 0.27$ | $3.92 \pm 0.19$ | $4.08 \pm 0.64$ | $2.89 \pm 0.29$ | $2.57 \pm 0.21$ | $2.37 \pm 0.37$ |
| Chlorophyll (AU) | $22.06 \pm 0.48$ | $21.70 \pm 0.57$ | $19.36 \pm 3.37$ | $18.37 \pm 2.90$ | $15.53 \pm 1.67$ | $19.85 \pm 2.72$ | $21.50 \pm 3.16$ | $18.61 \pm 2.3$ |
| Total sugar content (mg/g Glu eq.) | $5.39 \pm 0.08$ | $3.02 \pm 0.19$ | $3.29 \pm 0.68$ | $3.30 \pm 0.52$ | $3.52 \pm 0.62$ | $2.98 \pm 0.42$ | $3.41 \pm 0.42$ | $3.13 \pm 0.52$ |
| Phenolic index (ABS 320 nm/g) | $20.48 \pm 1.08$ | $23.41 \pm 0.44$ | $25.44 \pm 3.06$ | $22.16 \pm 2.97$ | $21.83 \pm 2.42$ | $23.20 \pm 2.10$ | $19.35 \pm 1.14$ | $25.39 \pm 3.0$ |
| Anthocyanins (mg/100 g) | $18.97 \pm 0.44$ | $23.31 \pm 0.31$ | $23.67 \pm 1.65$ | $22.51 \pm 1.36$ | $20.004 \pm 1.5$ | $21.99 \pm 1.54$ | $16.66 \pm 3.92$ | $22.74 \pm 1.6$ |
| Total chlorophyll $a + b$ (µg/mg) | $1.12 \pm 0.06$ | $1.52 \pm 0.005$ | $1.16 \pm 0.07$ | $1.36 \pm 0.08$ | $1.22 \pm 0.06$ | $1.02 \pm 0.03$ | $0.77 \pm 0.38$ | $1.043 \pm 0.0$ |
| Total carotenoids (µg/mg) | $0.15 \pm 0.02$ | $0.23 \pm 0.008$ | $0.18 \pm 0.009$ | $0.21 \pm 0.01$ | $0.20 \pm 0.01$ | $0.16 \pm 0.008$ | $0.11 \pm 0.05$ | $0.16 \pm 0.02$ |

Numbers represent total sugar content (mg/Kg Glu eq.), phenolic index (ABS 320 nm/g), anthocyanins (mg/100 g), total chlorophyll $a + b$ (µg/mg), total carotenoids (µg/mg) (n = 3), chlorophyll fluorescence (Fv/Fm, PI) and chlorophyll (AU) (n = 6). All data are presented as mean $\pm$ SE. When present, different letters represent significant differences ($p \leq 0.05$) among treatments and timepoints, calculated from a two-way ANOVA followed by Sidak's test.

The performance index (PI) and the maximum quantum efficiency of PSII (Fv/Fm) showed different trends during storage in the two species. Lettuce did not show a decline in Fv/Fm until the end of storage (Table 2). Rocket grown with both cultivation strategies showed a decline in Fv/Fm that reached values of around 0.81–0.82 after 9 days of storage (Table 3). The PI in lettuce increased, reaching values of 1.48–1.53 at the end of storage. Rocket leaves, instead, showed a progressive decline without a difference between the two pre-harvest treatments. At harvest, the PI values of rocket leaves were 3.86–4.13, and at the end of storage, they were almost halved (Table 3).

### 3.1.2. Non-Destructive Chlorophyll Content Estimation

Flushing did not affect leaf color (visual observation), and this was confirmed by the lack of noteworthy changes in the chlorophyll content, which was estimated in vivo at harvest (Table 1). Only in the case of lettuce did the chlorophyll content show higher values in the control compared to the flushing treatment when the treatment was applied 48 h before harvest (Table 1).

No changes were observed during storage for either species (Tables 2 and 3).

### 3.2. Evaluations Performed at Harvest and during Storage with Laboratory Analyses (Destructive Methods)

### 3.2.1. Total Sugar Content

Flushing treatments during cultivation produced no significant changes in sugars in either lettuce or rocket leaves at harvest (Tables 2 and 3).

During storage, sugar levels declined with time (Tables 2 and 3). Lettuce leaves treated with flushing 48 h before harvest did not show significant sugar changes. At the end of storage, the sugar concentration was higher in flushed lettuce (3.18 mg/g FW), while in the control, sugars were 2.75 mg/g FW (Table 2). However, in both cases, the changes observed were not significant.

### 3.2.2. Phenolic Index and Total Anthocyanins

The phenolic compound content (measured as the phenolic index) and the total anthocyanin concentration did not change in response to flushing treatment in any conditions at harvest (Table 1) or during storage in either species (Tables 2 and 3).

### 3.2.3. Chlorophyll (*a* + *b*) Content and Total Carotenoids

Chlorophyll (*a* + *b*) and total carotenoid content showed a similar trend between flushed and non-flushed leaves, indicating that visual and potential nutritional quality were not affected by the treatment in any of the tested conditions at harvest (Tables 1–3) or during storage (Tables 2 and 3). However, rocket leaves showed a chlorophyll reduction during storage independently of the growing conditions. The analysis of total chlorophyll content carried out in the laboratory confirmed the results obtained in vivo from the optical analysis.

### 3.2.4. Nitrate Concentration

Flushing treatment applied 24 h or 48 h before harvest was studied as a strategy for lowering nitrates. Flushing applied for 24 h significantly decreased nitrate concentration at harvest, with a reduction of 26% (Table 1).

Flushing treatments applied 48h before harvest further decreased the nitrate concentration by up to 56% (second trial—Table 1) and 42% (third trial—Table 2) in lettuce leaves at harvest (Tables 1 and 2). At harvest and during storage, lower values were observed in lettuce (Table 1), while in rocket, the changes in nitrate levels were not significant at any time point (Figure 1).

In rocket leaves subjected to the 48 h flushing treatment, the nitrate concentration decreased significantly during storage, with a reduction of 61% recorded at $T_2$ (Figure 1B).

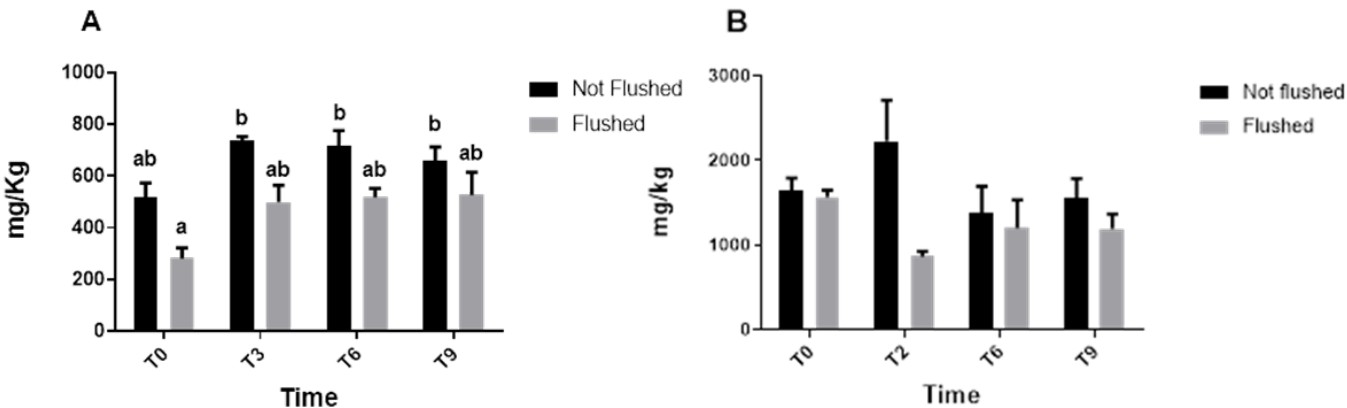

**Figure 1.** (**A**) Nitrate content measured in *Lactuca sativa* at harvest and three, six and nine days after harvest (data are mean $\pm$ SE, n = 4). (**B**) Nitrate content measured in *Eruca sativa* at harvest and two, six and nine days after harvest (data are mean $\pm$ SE, n = 4). Different letters represent significant differences ($p \leq 0.05$) among treatments and time points, calculated from a two-way ANOVA followed by Sidak's test.

### 3.3. *Weight Loss and Gas Content Determination in the Package Headspace*

### 3.3.1. Weight Loss

Lettuce leaves harvested from the 48 h flushing treatment maintained their weight during storage, while non-treated leaves showed higher weight loss percentages, especially and significantly at the end of storage (Figure 2A). After 9 days of storage, weight losses were double in non-flushed, which showed 2% weight loss compared to 1% observed in lettuce harvested from the flushed treatment. Both flushed and non-flushed rocket salad leaves increasingly lost weight, but not significantly, during storage (Figure 2B).

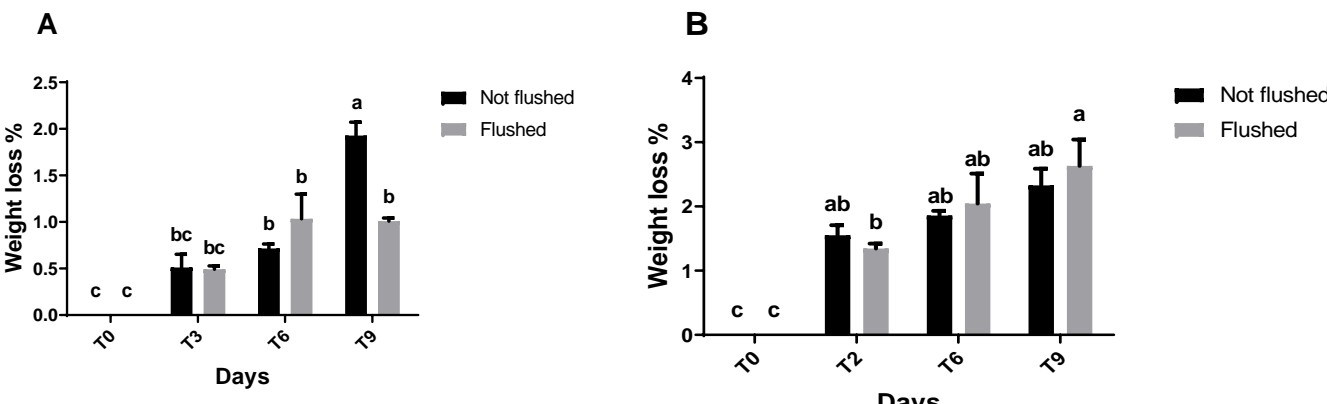

**Figure 2.** (**A**) *Lactuca sativa* weight loss (%) measured at harvest and three, six and nine days after harvest (data are mean ± SE, n = 4). (**B**) *Eruca sativa* weight loss (%) measured at harvest and two, six and nine days after harvest (data are mean ± SE, n = 3). Different letters represent significant differences ($p \leq 0.05$) among treatments and time points, calculated from a two-way ANOVA followed by Sidak's test.

### 3.3.2. $O_2$ Concentration

The $O_2$ concentration was higher in the lettuce without flushing treatment after three and six days of storage, while in flushed leaves, similar values were recorded during the entire storage duration (Figure 3A). However, there were no significant differences between treatments or among different storage times.

The $O_2$ concentration was only significantly higher in rocket with flushing treatment after six days of storage (Figure 3B).

### 3.3.3. $CO_2$ Concentration

The $CO_2$ content was higher in lettuce subjected to flushing treatment after three and six days of storage, while for leaves analyzed after nine days of storage, a higher content was recorded for the control samples (Figure 3A).

The $CO_2$ concentration was only significantly higher in rocket with flushing treatment after six days of storage (Figure 3B), while there were no significant differences between treatments or among storage times.

### 3.3.4. Relative Humidity (RH)

RH changes were observed inside the plastic punnets: control lettuce or rocket showed higher RH in punnets compared to those containing salads subjected to flushing (Figure 3).

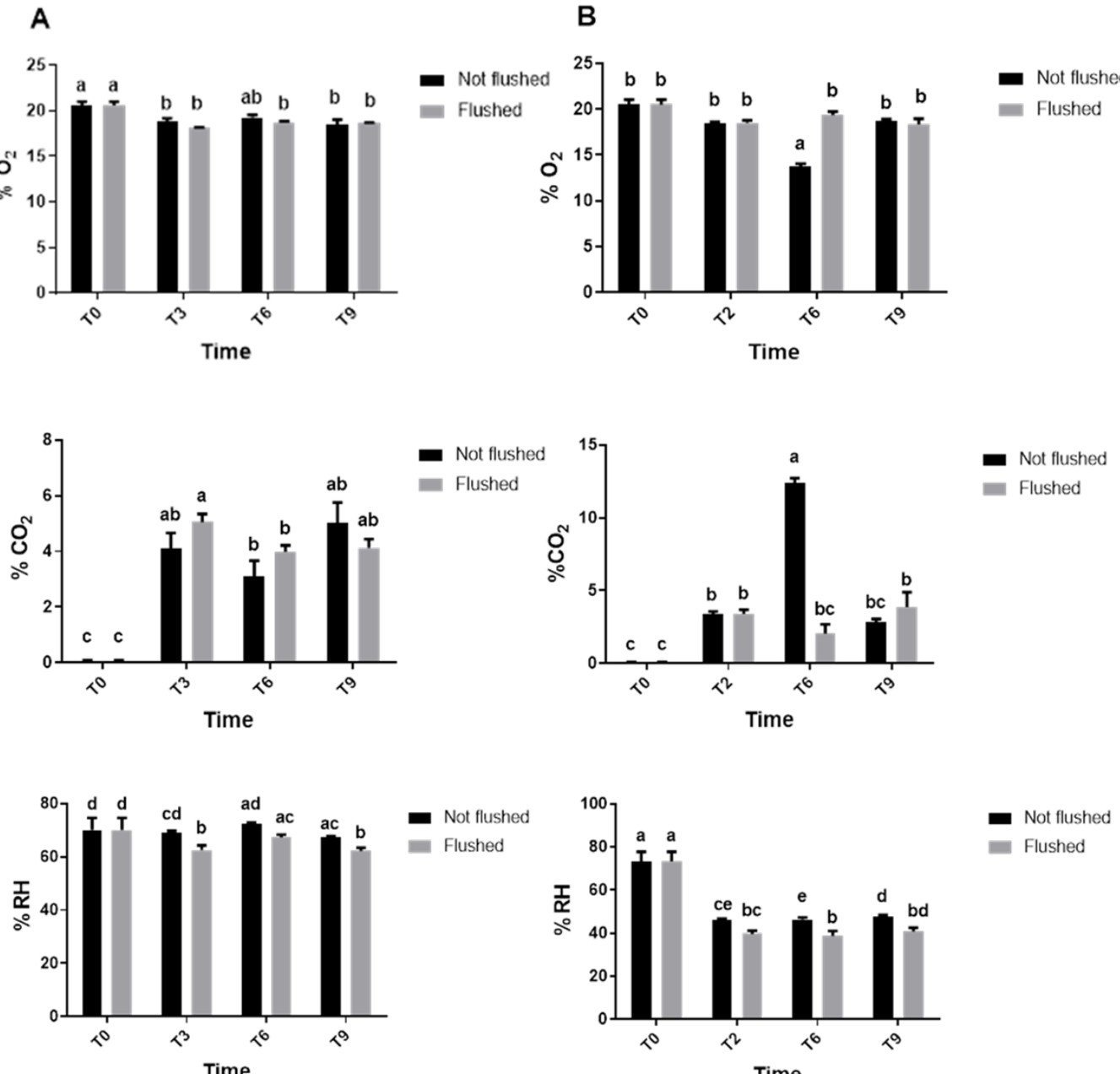

**Figure 3.** (**A**) $CO_2$, $O_2$ and RH concentrations inside lettuce (*Lactuca sativa*) plastic punnets measured at harvest and three, six and nine days after harvest (data are mean ± SE, n = 4). (**B**) $CO_2$, $O_2$ and RH concentrations inside *Eruca sativa* plastic punnets measured at harvest and three, six and nine days after harvest (data are mean ± SE, n = 3). Different letters represent significant differences ($p \leq 0.05$) among treatments and time points, calculated from 2-way ANOVA followed by Sidak's test.

## 4. Discussion

The quality of baby leaf vegetables depends on external and internal attributes. The visual appearance is determined by leaf color, which is mainly defined by chlorophyll, carotenoids, and anthocyanins [32]. Leaf color is the main quality parameter directly evaluated by the consumer when deciding to buy a product at the retailer. Chlorophyll pigments are also important for photosynthesis and light use efficiency in plants. Although some fluctuations in the chlorophyll index were observed, these changes were not significant; thus, it can be stated that the flushing treatment did not influence the accumulation of photosynthetic pigments and, therefore, did not change the leaf color (Tables 1–3). These

results showed that flushing did not affect the visual appearance and did not compromise the crop performance.

The maximum quantum efficiency of PSII is determined by the ratio between the variable fluorescence (Fv) and the fluorescence maximum (Fm) level when the plastoquinone electron acceptor (Qa) is totally reduced (Fm). This parameter is commonly considered a useful index to determine if a plant is subjected to stress conditions. In non-stressed herbaceous plants, the Fv/Fm values are usually greater than 0.83, while lower values indicate a foliar functionality loss [33]. The results from this study showed that the flushing treatment did not affect the chlorophyll a fluorescence-related parameters, and therefore, it can be deduced that the light system efficiency was maintained throughout the experiment in all trials (Tables 1–3), and the plants were in a good physiological state. Comparing the data collected with those of the other two experiments, where the photosynthetic performance related to supplemental light was investigated, similar data were obtained with values around 0.83 Fv/Fm [34,35].

The performance index is based on the leaf fluorescence measurement and reflects photosystem I and II functionality, providing quantitative information on the system performance under defined cultivation or storage conditions. Changes in the PI can also be affected by genetic differences among crops [26]. At harvest, stable values were recorded in all trials for both species considered in this experiment (Tables 1–3), while an increasing trend was observed during lettuce storage (Table 2). It can be hypothesized that the plant tissue optimizes resources during the postharvest phase as a physiological response in the adaptation process to different light and temperature conditions. In fact, the detached leaves were moved from a luminous cultivation environment with a temperature of 21–22 °C to a cold room at 8 °C in the dark. On the other hand, under the same conditions, we observed that rocket leaves were not able to react efficiently to different climatic and light conditions during the last storage period (Table 3).

Sugars serve as an energy reservoir as well as structural elements in plants. They also represent the main useful support for the plant respiratory reaction and a pivotal component contributing to the nutritional and sensory quality of the products [36]. There was a lower sugar content in lettuce leaves subjected to the flushing treatment when it was applied twice in the 24 h prior to harvesting (Table 1). On the contrary, in the other tests, an increase was observed with respect to the control, both at the harvest phase (Tables 1 and 2) and especially during storage (Table 2), where flushing was applied four times in the 48 h prior to harvesting. It could be assumed that this is due to a decrease in respiratory activity linked to weight loss and sugar level maintenance in the leaves. Compared to lettuce, there was a lower sugar content in rocket leaves subjected to the flushing treatment, both at harvest and during the storage phase (Table 3). It is possible that there was lower respiratory activity in non-flushed samples than in flushed ones. Overall, good sugar content was registered in lettuce and rocket leaves, indicating good photosynthetic activity during cultivation. The data obtained in this experiment are consistent with those published by other authors related to different growing conditions. In fact, similar or lower sugar values were registered as a response to different wavelength combinations [37] or in plants grown in an indoor hydroponic system under different light spectra [38].

Phenolic compounds, including anthocyanins, are secondary metabolites that play an important role in plant defense function against biotic and abiotic stresses (including UV light, insects and diseases). They can also confer characteristic color and sensory qualities to plant-derived food [39]. The analysis of the phenolic index and total anthocyanins showed that no significant variation was induced by the flushing treatment in any of the lettuce case studies, with slightly higher values when flushing was applied for 24 h (Table 1) and slightly lower values in samples subjected to 48 h flushing treatment (Tables 1 and 2) for both classes of antioxidants. Additionally, in rocket leaves, no significant variations were found at harvest or during storage (Table 3). This stability in bioactive compound content is an indication of how the product nutritional value was maintained under the various

conditions applied, without undergoing any stress, which would have probably led to significantly higher values than those detected.

Carotenoids are photosynthesis accessory pigments and chlorophyll-protective molecules that are able to promote light utilization and to prevent photooxidation. Carotenoids show antioxidant properties that help protect plants from oxidative stress by acting within the plant cells as well as along the energy transport chain [40]. Similar to what is reported for phenolic compounds, carotenoids also contribute to food products' nutraceutical quality due to their appreciated antioxidant and health-related properties. The carotenoid content measured in the present experiment was similar between treated and control leaves (Tables 1–3), indicating that the nutraceutical quality was not affected in any way by the treatment, either at harvest or during storage.

Nitrate concentration is a commercial quality parameter that must be controlled in leafy vegetables, and commercialization can be carried out respecting EU Regulation No. 1258/2011. The limitation of nitrate in leafy vegetables is related to the human diet and avoiding high daily intake that can be dangerous to human health [12]. The accumulation of nitrate in leaves depends on many factors, such as fertilization and nitrogen forms, light intensity, photoperiod, temperature, stress conditions and species [41,42]. Both lettuce and rocket are considered hyper-accumulators of nitrate, and, according to the current EU regulation, the control of $NO_3$ levels in these species is crucial [43]. As we proposed, a simple way to reduce nitrate accumulation in the commercial product can be the substitution of the nutrient solution with tap water before harvesting (flushing) [44]. Flushing treatments applied 48 h before harvest in lettuce (Table 1 and Figure 1A) and rocket (Figure 1B) represent a cheap strategy to lower the nitrate concentration at harvest. Changes in nitrate content during storage have been observed in lettuce from harvest (T0) to the first time point (T3) and may be due to a decrease in the water content, which probably increased the concentration of $NO_3$ (not because of an increase in its accumulation). Numerous studies have been conducted aiming to identify valid approaches to efficiently reduce the nitrate content. In conventional and integrated agriculture, higher nitrate values were recorded in lettuce leaves, while biological agriculture showed similar values to the following case studies [45]. Higher nitrate values were recorded in rocket leaves subjected to flushing treatment 48 h before harvesting in an indoor hydroponic system or a floating system [42–46].

From the analyses of gas concentrations in the PP punnet headspace during storage, lower relative humidity (RH) values were recorded in the cases of lettuce (Figure 3A) and rocket (Figure 3B) subjected to flushing compared to non-flushed control samples. Tissue respiration appears to be the main cause of the weight reduction and the decrease in the product quality during the storage process. Slightly reduced respiration was observed in the flushed leaves compared to those not subjected to any treatment (Figure 3A,B). It was noted that at the end of storage, the $CO_2$ level accumulated inside the lettuce packages was higher in the control samples (Figure 3A), indicating a slightly faster respiratory rate. In rocket packages, no significant differences were observed between the two treatments at two and nine days after the harvest phase (Figure 3B). After six days, a higher $CO_2$ level was observed in the untreated samples compared to the flushed ones and the other controls at the other times (Figure 3B); this did not significantly affect the sugar or nitrate contents. It can be hypothesized that the fluctuations observed in the gas levels within the punnets at different storage times were probably due to imperfect hermetic plastic sealing. The $O_2$ concentration showed similar values and was constant over time for both treatments and cultures (Figure 3A,B), except six days after harvest, where lower values were recorded for the untreated rocket salad samples (Figure 3B) in relation to the $CO_2$ content mentioned above. The weight reduction (Figure 1A,B) is also due to the water loss due to vegetable tissue transpiration, thus leading to an accumulation of humidity inside the package. Good oxygenation and low relative humidity levels prevented, during the storage phase, nitrite formation from the nitrate present at harvest.

## 5. Conclusions

In conclusion, from the results obtained, it can be stated that it is possible to reduce the nitrate concentration in baby leaf lettuce and rocket using the flushing treatment. In all of the trials performed, no stress conditions or yield loss occurred, and the quality and nutritional parameters were not significantly influenced by the treatment, either at harvest time or during storage. Furthermore, it should be considered that the flushing treatment application was more efficient when applied starting 48 h before harvest, as the lowest nitrate concentration was recorded. It was observed that the flushing treatment was more effective in lettuce (Table 1 and Figure 1A) than in rocket at harvest (Figure 1B), indicating the possibility of further optimizing this procedure for a specific species. This treatment, together with an optimized controlled cultivation system, can be a valid alternative to other strategies already tested in conventional and integrated cultivation systems. The flushing treatment is therefore an innovative, efficient and sustainable strategy compared to other nitrate management systems.

**Supplementary Materials:** The following supporting information can be downloaded at: https://www.mdpi.com/article/10.3390/horticulturae8070604/s1, Table S1: detailed results related to the ANOVA analyses.

**Author Contributions:** G.C., B.M.F. and A.F.: conceptualization and methodology; D.G. and G.C.: formal analysis; D.G. and G.C.: data curation; D.G.: writing—original draft preparation; G.C., A.F. and B.M.F.: writing—review and editing; G.C. and A.F.: supervision. All authors have read and agreed to the published version of the manuscript.

**Funding:** This research received no external funding.

**Conflicts of Interest:** The author Benjamin Franchetti is employed by the company Agricola Moderna, Sunspring Srl. All other authors declare no competing interest.

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
