# Peer review of "The Effect of Flushing on the Nitrate Content and Postharvest Quality of Lettuce (Lactuca sativa L. Var. Acephala) and Rocket (Eruca sativa Mill.) Grown in a Vertical Farm"

_horticulturae, doi:10.3390/horticulturae8070604_

Round 1
Reviewer 1 Report
The corrections specified on the manuscript must be completed.

Author Response
All the changes and corrections have been made. Specific replies to the comments are reported into the pdf file.

Reviewer 2 Report
In my opinion, the area of research related to the article is relevant. However, I think that the material and methods section needs to be improved.
The text states “The results obtained indicate that is possible to reduce the nitrates concentration, by up to 56% in lettuce and 61% in rocket, while maintaining the products quality in baby leaves, by an optimized management of the nutrient solution.
To me, this sentence is not clear. Was the nutrient solution optimized, or replaced by water?
Was there a reduction of electrical conductivity by adding water or a completely replacing of the nutrient solution with water? To me, that is not clear. Can you better clarify this in the material and methods?
Line 40 - “Inert substrates, such as peat, perlite, vermiculite, rock wool and zeolite are used as soil substitutes to provide support to the roots [4]”.
Peat is not an inert substrate.
Line 93 I suggest replacing “fertirrigation” by fertigation
Line 151 The nutrient solution consisted of a modified Hoagland’s solution: the nutrients concentrations expressed as mM, were: 12 N-NO3, 3.8 N-NH4, 2.8 P, 8.4 K, 3.5 152 Ca, 1.4 Mg and Hoagland’s concentration for micronutrients.
Did the nutrient solution have no sulfur?
Line 157 - In the first trial, the flushing treatment was applied twice, 24 h before harvesting, with an interval of 12 h.
To me, that is not clear. Was there applied nutrient solution between the flushing? How was the water or the nutrient solution applied? On the other hand, it was also important to know the amount of water applied, as this can reduce the nutrients in the substrate to a greater or lesser extent. What was the volume or dimensions of the trays?
Line 159- Had the experiment statistical design? How many plants per treatment?
Figure 1. A. On the x-axis I suggest replacing time with days
To me, that is not clear. Was applied nutrient solution between the flushing. Was it only applied water in the last 24 hours before harvest? How was the water or the nutrient solution applied? On the other hand, it was also important to know the amount of water applied, as this can reduce the nutrients in the substrate to a greater or lesser extent.
Author Response
Reviewer # 2
Comments and Suggestions for Authors
In my opinion, the area of research related to the article is relevant. However, I think that the material and methods section needs to be improved. The text states “The results obtained indicate that is possible to reduce the nitrates concentration, by up to 56% in lettuce and 61% in rocket, while maintaining the products quality in baby leaves, by an optimized management of the nutrient solution.
To me, this sentence is not clear. Was the nutrient solution optimized, or replaced by water?
Authors answer (A.A.): We agree with the reviewer. The sentence has been changed according to this suggestion. We hope that it would be clearer in this way.
Was there a reduction of electrical conductivity by adding water or a completely replacing of the nutrient solution with water? To me, that is not clear. Can you better clarify this in the material and methods?
Authors answer (A.A.): Since it can be easily applied in commercial cultivation systems, in this trial the nutrient solution was completely replaced by water. In the introduction section we wanted to explain that there are two options (complete replacement or partial water addition). We clearly stated that in these trials the replacement was complete (we changed the sentence in the abstract as well as in the M&M section). We hope that in this way it would be clearer for the readers.
Line 40 - “Inert substrates, such as peat, perlite, vermiculite, rock wool and zeolite are used as soil substitutes to provide support to the roots [4]”.
Peat is not an inert substrate.
Authors answer (A.A.): we are sorry for the mistake. We changed the sentence, referring to those substrates as “growing” substrates.
Line 93 I suggest replacing “fertirrigation” by fertigation
Authors answer (A.A.): changed accordingly.
Line 151 The nutrient solution consisted of a modified Hoagland’s solution: the nutrients concentrations expressed as mM, were: 12 N-NO3, 3.8 N-NH4, 2.8 P, 8.4 K, 3.5 152 Ca, 1.4 Mg and Hoagland’s concentration for micronutrients.
Did the nutrient solution have no sulfur?
Authors answer (A.A.): the reviewer is right; we missed this information. The S concentration was 3.5 mM. We added this information into the text.
Line 157 - In the first trial, the flushing treatment was applied twice, 24 h before harvesting, with an interval of 12 h.
To me, that is not clear. Was there applied nutrient solution between the flushing?
Authors answer (A.A.): yes, the nutrient solution was applied between the flushing. We specified this aspect into the text.
Was it only applied water in the last 24 hours before harvest?
Authors answer (A.A.): The flushing treatment was applied as described 24 or 48 hours before harvesting. This strategy allows the reduction of nitrate concentration without affecting the overall quality of produce.
How was the water or the nutrient solution applied?
Authors answer (A.A.): The cultivation system was the one normally used for the company purposes, and it is basically an ebb-and-flow hydroponic system, with automatic management of the nutrient solution. In this case one tank normally used for the nutrient solution was filled with water and the system was programmed according to the experimental conditions (number of flushing ecc...).
On the other hand, it was also important to know the amount of water applied, as this can reduce the nutrients in the substrate to a greater or lesser extent. What was the volume or dimensions of the trays?
Authors answer (A.A.): the volume of the recirculating nutrient solution (or water) was around 0.5 L per m2 and the nutrient solution (or water in case of flushing) was 0.5 L/min.
Line 159- Had the experiment statistical design? How many plants per treatment?
Authors answer (A.A.): for each species, treatment was applied on four trays (30x26 cm). All the analytical determinations, were conducted on randomly selected leaves, chosen among 128 plants in the first trial, 144 plants in the second one, 288 plants in the third one and 144 rocket plants, for both treatments (flushing and control). After harvest and during storage, lettuce and rocket leaves were sampled in quadruplicate.
Figure 1. A. On the x-axis I suggest replacing time with days
Authors answer (A.A.): Figure 1 (now Figure 2) has been modified according to this suggestion.

Reviewer 3 Report
MS No. horticulturae-1783193 (Guffanti et al.) summarizes the results of a hydroponics study with Lactuca sativa (lettuce) and Eruca sativa (rocket), that aimed to test whether pre-harvest replacement of the nutrient solution with water would reduce the nitrate content and affect the quality of the harvested leafy vegetables. The Authors used different durations of water flushing and measured several physiological and quality parameters both before and after the harvest. They summarized that pre-treatment with water can be an applicable strategy to reduce the nitrate content of the biomass while still preserving its quality and shelf-life.
In general, the manuscript concerns a recent topic of food production, therefore it fits into the scope of the journal; its aims are clear and the applied methods are suitable to address them. There are, on the other hand, several flaws that need to be elaborated on before its acceptance for publication. Double-checking the typos, spelling and grammar would also be advisable.
Specific remarks:
Table 3, Figures 2 and 3: the lower halves of the table/figures went missing during the conversion of the manuscript into pdf. The Authors should resubmit the manuscript and overcome somehow this issue.
l 150: please specify the spectral properties of the applied irradiation (either the type of LEDs or the spectral composition)
l 154: please specify if it was tap water or deionized water. In the former case, was the NO3-content and other water chemical and physical parameters checked?
l 157-159: based on the description, it’s not clear how the flushing with water was performed. Was it a continuous supply with water instead of the nutrient medium? What do the 12 h intervals mean?
l 203: “1 g of leaf tissue (around 2 leaves) was ground in 3 mL and 4 mL distilled water” – Does 3 and 4 mL apply to lettuce and rocket? Please state it clearly.
Results: I would suggest you to reorder this section, and separate the results into two major subsections, based on whether they present the data for pre-harvest treatments or post-harvest storage.
Throughout the Results and Discussion sections: as you applied two factors -time and treatment, respectively- and performed two-way ANOVAs, you should report the ANOVA summary tables, and check if either factor had significant effect on the given parameter. You can state significant or non-significant effects only if those were confirmed by the ANOVA, and can deal with the results of post hoc pairwise comparisons following that. E.g. you state that sugar concentrations did not change significantly during storage in Lactuca sativa even though the sugar content halved during the storage in the control according to Table 2. Also, you say that “At harvest and during the storage lower values were observed in the lettuce and rocket subjected to flushing treatment (Table 1 - Figure2)“, though based on Figure 2.A there were no significant differences in either species due to the pre-treatment comparing at any given time point.
l 295: “between the two storage conditions” – what are the two storage conditions?
Tables 1, 2 and 3: you should omit the titles (e.g. “Lactuca sativa) from above the tables, as the captions report the necessary info. Also, you don’t need to indicate the units in the footnotes as they are indicated in the tables themselves.
l 362-363: “In the storage phase sugars declined with the storage time and the concentration” –concentration of what?
l 391: Table 2.A?
l 490-495: You can omit or reduce this part as it does not belong strictly to the topic.
Author Response
Reviewer # 3
Comments and Suggestions for Authors
MS No. horticulturae-1783193 (Guffanti et al.) summarizes the results of a hydroponics study with Lactuca sativa (lettuce) and Eruca sativa (rocket), that aimed to test whether pre-harvest replacement of the nutrient solution with water would reduce the nitrate content and affect the quality of the harvested leafy vegetables. The Authors used different durations of water flushing and measured several physiological and quality parameters both before and after the harvest. They summarized that pre-treatment with water can be an applicable strategy to reduce the nitrate content of the biomass while still preserving its quality and shelf-life.
In general, the manuscript concerns a recent topic of food production, therefore it fits into the scope of the journal; its aims are clear and the applied methods are suitable to address them. There are, on the other hand, several flaws that need to be elaborated on before its acceptance for publication. Double-checking the typos, spelling and grammar would also be advisable.
Specific remarks:
Table 3, Figures 2 and 3: the lower halves of the table/figures went missing during the conversion of the manuscript into pdf. The Authors should resubmit the manuscript and overcome somehow this issue.
Authors answer (A.A.): thank for the comment. After the acceptance, we will provide full size hq images files for each figure, to be used in the preparation of the final version of the manuscript.
l 150: please specify the spectral properties of the applied irradiation (either the type of LEDs or the spectral composition)
Authors answer (A.A.): lighting was provided by Valoya, 28 W PAR LED lamps. This information has been included into the main text of the manuscript.
l 154: please specify if it was tap water or deionized water. In the former case, was the NO3-content and other water chemical and physical parameters checked?
Authors answer (A.A.): in order to describe a real productive context, we used tap water. The main characteristics of the water used are reported in the following table. Data are available at the following link: https://www.latuaacqua.it/wps/portal/latuaacqua/it/home/!ut/p/z1/04_Sj9CPykssy0xPLMnMz0vMAfIjo8zifQ0sjdwtTIx83f2DnQwCjQ0Dg92MTA28A0z1wwkpiAJKG-AAjgZA_VFgJThNMIEqwGNGQW6EQaajoiIAyntlvg!!/dz/d5/L2dBISEvZ0FBIS9nQSEh/:
Parameter |
law limits |
Milan water |
Units |
|
Ammonium |
0,5 |
<0.10 |
mg/l |
|
Arsenic |
10 |
<2 |
µg/l |
|
Bicarbonate |
214 |
mg/l |
||
Calcium |
84 |
mg/l |
||
Chloride |
0.2* |
<0.01 |
mg/l |
|
chloride ions |
250 |
31 |
mg/l |
|
EC |
2500 |
571 |
µS/cm a 20° |
|
Durezza |
15-50 |
28 |
°F |
|
Fluoride |
1,5 |
<0.5 |
mg/l |
|
Magnesium |
17,3 |
mg/l |
||
Manganese |
50 |
<1 |
µg/l |
|
Nitrate |
50 |
24 |
mg/l |
|
Nitrate |
0,5 |
<0.20 |
mg/l |
|
pH |
Da 6.5 a 9.5 |
7,6 |
Unità pH |
|
Potassium |
10 |
2 |
mg/l |
|
dry matter |
1500* |
408 |
mg/l |
|
Sodium |
200 |
19 |
mg/l |
|
sulfur |
250 |
50 |
mg/l |
l 157-159: based on the description, it’s not clear how the flushing with water was performed. Was it a continuous supply with water instead of the nutrient medium? What do the 12 h intervals mean?
Authors answer (A.A.): in this trial the nutrient solution was completely replaced by water, which was continuously supplied. During the intervals between flushing treatments, the nutrient solution was administered regularly. The volume of the recirculating nutrient solution (or water) was around 0.5 L per m2 and the nutrient solution (or water in case of flushing) was 0.5 L/min.
l 203: “1 g of leaf tissue (around 2 leaves) was ground in 3 mL and 4 mL distilled water” – Does 3 and 4 mL apply to lettuce and rocket? Please state it clearly.
Authors answer (A.A.): the reviewer is right; the sentence has been changed according to this comment.
Results: I would suggest you to reorder this section, and separate the results into two major subsections, based on whether they present the data for pre-harvest treatments or post-harvest storage.
Authors answer (A.A.): we agree with the reviewer, however we would prefer maintaining the pre- and post-harvest description of the results as a whole text. The main reason is that data collected at harvest represent the starting point in the postharvest evaluation.
To follow the reviewer suggestion, we tried to better divide the description of the results related to the two phases, with more emphasis on this aspect.
Also, we moved figure 1 (related to postharvest weigh loss) to the final part of the results section, together with other postharvest-related determinations.
We hope that these changes will help the reader and make the manuscript clearer.
Throughout the Results and Discussion sections: as you applied two factors -time and treatment, respectively- and performed two-way ANOVAs, you should report the ANOVA summary tables, and check if either factor had significant effect on the given parameter.
Authors answer (A.A.): detailed tables reporting the results for each ANOVA analysis are reported in the new Supplementary file 1.
You can state significant or non-significant effects only if those were confirmed by the ANOVA, and can deal with the results of post hoc pairwise comparisons following that. E.g. you state that sugar concentrations did not change significantly during storage in Lactuca sativa even though the sugar content halved during the storage in the control according to Table 2.
Authors answer (A.A.): We checked all the results and, about total sugars in lettuce, we confirm that no significant changes were found, and no effect or significant interaction was found for time or treatment.
Also, you say that “At harvest and during the storage lower values were observed in the lettuce and rocket subjected to flushing treatment (Table 1 - Figure2)“, though based on Figure 2.A there were no significant differences in either species due to the pre-treatment comparing at any given time point.
Authors answer (A.A.): The reviewer is right; the differences observed were significant only for lettuce. We revised the text accordingly.
l 295: “between the two storage conditions” – what are the two storage conditions?
Authors answer (A.A.): there was a mistake. The sentence has been modified and reference to pre-
harvest conditions have been added.
Tables 1, 2 and 3: you should omit the titles (e.g. “Lactuca sativa) from above the tables, as the captions report the necessary info. Also, you don’t need to indicate the units in the footnotes as they are indicated in the tables themselves.
Authors answer (A.A.): Tables have been modified according to the reviewer suggestion.
l 362-363: “In the storage phase sugars declined with the storage time and the concentration” –concentration of what?
Authors answer (A.A.): the sentence has been changed
l 391: Table 2.A?
Authors answer (A.A.): “2.A” changed to “2”.
l 490-495: You can omit or reduce this part as it does not belong strictly to the topic.
Authors answer (A.A.): we agree with this comment. This part has been deleted.

Round 2
Reviewer 1 Report
Manuscripts can be accepted as corrected.
Author Response
Dear Editor,
Once again thank you for your consideration, we are submitting a revised version of our manuscript entitled: “The effect of flushing on the nitrate content and postharvest quality of lettuce (Lactuca sativa L. Var. Acephala) and rocket (Eruca sativa Mill.) grown in a vertical farm”.
We carefully addressed the minor issues raised in the latest comments and we hope that the Editorial Board will agree that our manuscript may be acceptable for publication in Horticulturae.
All changes made in the manuscript are marked using track changes.
Giacomo Cocetta, on behalf of the authors.
Reviewer # 1
Manuscripts can be accepted as corrected.
Authors answer: we thank the reviewer for the valuable help and for all the suggestions.
Reviewer # 2
The manuscript was improved.
Authors answer: we thank the reviewer for the valuable help and for all the suggestions.
Reviewer # 3
The Authors have modified the manuscript according to the reviewers’ remarks. Some further spell- and grammar check would be advisable.
Authors answer: We thank the reviewer for the valuable help and suggestions. We further revised the whole manuscript and fixed some more spell- and grammar errors.
Also, an indication that tap water was used is still missing from the M&M section.
Authors answer: this information has been added into the M&M section.
In addition, the explanation for the higher CO2 and lower O2 concentrations at the 6th day of storage in rocket due to imperfect sealing (l 620-627) seems a bit vague, as one could expect the opposite, that is lower CO2 and higher O2, if the gas exchange was more efficient between the punnets and their environment.
Authors answer: the reviewer is right. We changed the sentence to: “It can be hypothesized that the fluctuations observed in the gases level within the punnets at different storage times, was probably due to not perfectly hermetic plastic sealing”. Hopefully, in this way the explanation would be clearer for the readers.
Reviewer 2 Report
The manuscript was improved.
Author Response

(The authors gave the same response as above.)

Reviewer 3 Report
The Authors have modified the manuscript according to the reviewers’ remarks. Some further spell- and grammar check would be advisable. Also, an indication that tap water was used is still missing from the M&M section.
In addition, the explanation for the higher CO2 and lower O2 concentrations at the 6th day of storage in rocket due to imperfect sealing (l 620-627) seems a bit vague, as one could expect the opposite, that is lower CO2 and higher O2, if the gas exchange was more efficient between the punnets and their environment.
Author Response
Dear Editor,
Once again thank you for your consideration, we are submitting a revised version of our manuscript entitled: “The effect of flushing on the nitrate content and postharvest quality of lettuce (Lactuca sativa L. Var. Acephala) and rocket (Eruca sativa Mill.) grown in a vertical farm”.
We carefully addressed the minor issues raised in the latest comments and we hope that the Editorial Board will agree that our manuscript may be acceptable for publication in Horticulturae.
All changes made in the manuscript are marked using track changes.
Giacomo Cocetta, on behalf of the authors.
Reviewer # 1
Manuscripts can be accepted as corrected.
Authors answer: we thank the reviewer for the valuable help and for all the suggestions.
Reviewer # 2
The manuscript was improved.
Authors answer: we thank the reviewer for the valuable help and for all the suggestions.
Reviewer # 3
The Authors have modified the manuscript according to the reviewers’ remarks. Some further spell- and grammar check would be advisable.
Authors answer: We thank the reviewer for the valuable help and suggestions. We further revised the whole manuscript and fixed some more spell- and grammar errors.
Also, an indication that tap water was used is still missing from the M&M section.
Authors answer: this information has been added into the M&M section.
In addition, the explanation for the higher CO2 and lower O2 concentrations at the 6th day of storage in rocket due to imperfect sealing (l 620-627) seems a bit vague, as one could expect the opposite, that is lower CO2 and higher O2, if the gas exchange was more efficient between the punnets and their environment.
Authors answer: the reviewer is right. We changed the sentence to: “It can be hypothesized that the fluctuations observed in the gases level within the punnets at different storage times, was probably due to not perfectly hermetic plastic sealing”. Hopefully, in this way the explanation would be clearer for the readers.
This manuscript is a resubmission of an earlier submission. The following is a list of the peer review reports and author responses from that submission.
Round 1
Reviewer 1 Report
In my opinion, the apparent conclusion of this study is that the nutrient solution (NS) that was used to grow baby-leaf lettuce and rocket plants was extremely rich in N-NO3 (12mM). Provided that there was no adverse effect in yield or physiological parameters, the plants probably over-accumulated all minerals and nitrates, as well, losing any ability to further absorb any element during the last 48 hours before harvest. This is also confirmed by the fact that even when the whole NS was replaced by water, no significant effect was exhibited. It would be more rational to exclude only nitrates for the ‘flushing’ solution, to avoid nitrates accumulation. The authors should better determine the actual needs of the crops in terms of nutrient supply before totally removing any supply from the plants.
Moreover, the number of replications should be substantially increased. A number of reps such as n=2 by no means can generate solid conclusions and this is also conformed by the variation that was observed within the treatments among the storage days.
In addition, the storage period should be included in the analysis of variance of the data, in order to assess the effect of storage on each variable and test whether any ‘increases’ or ‘decreases’ during storage were observed.
Some other minor comments:
L86-89: This is in contrast with the sentence above in L81-86.
L511: How would nitrate accumulate postharvest , as long as the roots are detached for the nitrogen source, namely the nutrient solution?
Author Response
Reviewer 1
In my opinion, the apparent conclusion of this study is that the nutrient solution (NS) that was used to grow baby-leaf lettuce and rocket plants was extremely rich in N-NO3 (12mM). Provided that there was no adverse effect in yield or physiological parameters, the plants probably over-accumulated all minerals and nitrates, as well, losing any ability to further absorb any element during the last 48 hours before harvest. This is also confirmed by the fact that even when the whole NS was replaced by water, no significant effect was exhibited. It would be more rational to exclude only nitrates for the ‘flushing’ solution, to avoid nitrates accumulation. The authors should better determine the actual needs of the crops in terms of nutrient supply before totally removing any supply from the plants.
Authors response (A.R.): We thank the reviewer for this comment. This is a simulation of practical application in a baby leaf production in a vertical farm. In production systems the nutrient solution concentrations used are higher in order to provide a buffer of nutrients for all growing cycle that is about 25-35 days. Nutrient solutions for commercial hydroponic systems range from 10-15 mM N. The first nutrient solution for commercial purpose is the Hoagland nutrient solution, which was first made in the 1940. This has been optimised for different crops see references, Sonneveld, C. (1985). A method for calculating the composition of nutrient solutions for soilless cultures (No. 57). Glasshouse Crops Research and Experiment Station, Sonneveld, C., & Voogt, W. (2009). Nutrient solutions for soilless cultures. In Plant nutrition of greenhouse crops (pp. 257-275). Springer, Dordrecht.
The amount of nitrate during the early stage of vegetables cultivation allows fast growing and shorter cycles.
In commercial production system is not convenient to have addition tanks with nutrient solution without the nitrate since this nutrient solution should be only used 1-2 days before harvest. Moreover, the nutrient solution without nitrate or ammonium can generate unbalance in nutrient uptake. The cheapest way is to close the nutrient solution supply network and flush the plants with only water. This strategy allows the utilization of nitrate that is accumulated in the vacuoles.
Moreover, the number of replications should be substantially increased. A number of reps such as n=2 by no means can generate solid conclusions and this is also conformed by the variation that was observed within the treatments among the storage days.
A.R.: The only dataset with N=2 (referring to yield) has been removed, it was reported because the harvest of each tank was split in two in order to have a better evaluation of the variability. Detailed information regarding the number of replications used for each analysis are reported into the manuscript.
In addition, the storage period should be included in the analysis of variance of the data, in order to assess the effect of storage on each variable and test whether any ‘increases’ or ‘decreases’ during storage were observed.
A.R.: All data have been analysed again, including the multiple comparison and differences among timepoints and treatments.
Some other minor comments:
L86-89: This is in contrast with the sentence above in L81-86.
A.R.: We change the sentences, avoiding contrasting information and reporting a more appropriate reference to the current literature.
L511: How would nitrate accumulate postharvest, as long as the roots are detached for the nitrogen source, namely the nutrient solution?
A.R.: We agree with this comment. Changes in nitrate content during storage have been observed in lettuce from harvest (T0) to the first time point (T3) and are due to a decrement in the water content, which have probably increased the concentration of NO3 (not because of an increment in its accumulation). No significant changes were observed in rocket. These considerations have been added into the manuscript.

Reviewer 2 Report
Generally, this manuscript needs to be large improved and discussed in more details before published.
The introduction needs to be improved according to the main aim of this study. In the introduction part the quality parameter aspects such as Fv/Fm, Chlorophy II (AU), total carotenoids etc. could be explained in details. The references related to other researches as basements of this research work need to be listed. Although in discuss these were discussed, but it cannot be replaced. The introduction and discuss need to be co-related.
In materials and methods, the experiment design needs to be explained in details rather than describe the methods as already published. For example, how many replicates? how the plants were harvested, and the randomly. The statistical methods need to be described more in details.
Results, tables need to be more refined. In table 3 the ChlorophyII was 15.53±1,67 for T6, but for T9, this value was 21.50±3.16. These kinds of values needed to be explained. Fig 3. CO2% T6 was extremely high. The reasons needed to be discussed.
In discuss, this part was hard to read. The data compared could be list sources, eg in which table? The results could be distinguish from other research work or from the present study.
In text there are several empty spaces. And the grammar need to be checked in details.
Author Response
Reviewer 2
The introduction needs to be improved according to the main aim of this study. In the introduction part the quality parameter aspects such as Fv/Fm, Chlorophy II (AU), total carotenoids etc. could be explained in details. The references related to other researches as basements of this research work need to be listed. Although in discuss these were discussed, but it cannot be replaced. The introduction and discuss need to be co-related.
A.R.: As suggested by the reviewer, more information related to quality aspects (including, sugars, leaf pigments, and chlorophyll a fluorescence) have been added to improve the quality of the introduction.
In materials and methods, the experiment design needs to be explained in details rather than describe the methods as already published. For example, how many replicates? how the plants were harvested, and the randomly. The statistical methods need to be described more in details.
A.R.: More information has been added into the text to better explain the experimental design and the analytical procedures.
Results, tables need to be more refined. In table 3 the ChlorophyII was 15.53±1,67 for T6, but for T9, this value was 21.50±3.16. These kinds of values needed to be explained.
A.R.: We agree with this comment about fluctuations in the chlorophyll index during storage. However, considering the results from the statistical analysis, we conclude that the chlorophyll content was not affected by treatments. We added these considerations into the discussion.
Fig 3. CO2% T6 was extremely high. The reasons needed to be discussed.
A.R.: We noticed that this data was quite high since we performed the experiments and analyses. We hypothesized that the plastic punnets that we used (which are commercially available and commonly used for postharvest storage of fruit and vegetables) were not perfectly hermetic and did not offer the same barrier among different samples (especially considering the closure of the cap). We already mentioned this aspect in the discussion section.
In discuss, this part was hard to read. The data compared could be list sources, eg in which table? The results could be distinguish from other research work or from the present study. In text there are several empty spaces. And the grammar need to be checked in details.
A.R.: The whole discussion section has been revised and rephrased following the reviewer suggestions. The whole manuscript has been carefully revised and grammar editing has been performed.

Reviewer 3 Report
The manuscript is relevant and deals with a strategy that can reduce the accumulation of nitrate without harming the commercial characteristics of lettuce and rocked salad.
Parts of the text have very long paragraphs. There are suggestions marked in the text to break the paragraph and start a new paragraph.
The methodology is confused as to what the experiment or treatment actually was.
The methodology of some variables is incomplete.
The results lead to the understanding that they are presented jointly for the three experiments with lettuce and if that is indeed the case, it is evident that what was called an experiment are actually treatments.
The discussion and conclusions are very good.

Author Response
Reviewer 3
The manuscript is relevant and deals with a strategy that can reduce the accumulation of nitrate without harming the commercial characteristics of lettuce and rocked salad.
Parts of the text have very long paragraphs. There are suggestions marked in the text to break the paragraph and start a new paragraph.
A.R.: We thank the reviewer for the valuable help. We change the manuscript structure according to your suggestions.
The methodology is confused as to what the experiment or treatment actually was.
A.R.: Four trials were carried out at different times: the first one on lettuce with control and flushing treatment 24 hours; the second one on lettuce with control and flushing treatment 24 and 48 hours; the third one on lettuce with control and flushing treatment 24 and 48 hours with also the storage phase. The last trial was conducted on rocket salad with control and flushing treatment 48 hours with also the storage phase. The results were discussed together because the experimental plan was applied in the same way and the trials were combined with the same analytical procedures and the same treatments, obtaining similar results on the treatment effectiveness both between the trials on lettuce and then on rocket salad.
We tried to improve the M&M section by adding more details and information.
The methodology of some variables is incomplete.
A.R.: More information has been added into the text to better explain the experimental design and the analytical procedures.
The results lead to the understanding that they are presented jointly for the three experiments with lettuce and if that is indeed the case, it is evident that what was called an experiment are actually treatments.
A.R.: We tried to better explain the structure of the whole experiment, indicating the number of trials and how they were performed. We hope that in this way the experimental plant would be cleared for the readers.
The discussion and conclusions are very good.
A.R.: Thank you very much for the positive response. According to the other reviewers’ comments, the whole discussion section has been revised and rephrased to further improve it.

Round 2
Reviewer 1 Report
The authors provided sufficient explanations to the comments and have submited a revised manuscript taking into account the reviewers' recommendations.
Reviewer 2 Report
The grammar still need to be improved.
eg. Latin name should be in italic.
"rocket salad" or "rocket"?